# Expansion of social protection is necessary towards zero catastrophic costs due to TB: The first national TB patient cost survey in the Philippines

Jhiedon L. Florentino[1], Rosa Mia L. Arao[1]*, Anna Marie Celina Garfin[2], Donna Mae G. Gaviola[2], Carlos R. Tan[1], Rajendra Prasad Yadav[3], Tom Hiatt[3], Fukushi Morishita[4], Andrew Siroka[5], Takuya Yamanaka[5,6,7], Nobuyuki Nishikiori[5]

1 Health Policy Development Program (HPDP)-UPecon Foundation, Inc., Quezon City, Philippines,
2 National TB Control Programme, Department of Health, Manila, Philippines, 3 World Health Organization, Country Office, Manila, Philippines, 4 World Health Organization, Regional Office for the Western Pacific, Manila, Philippines, 5 World Health Organization, Global Tuberculosis Programme, Geneva, Switzerland, 6 Department of Global Health and Development, London School of Hygiene and Tropical Medicine, London, United Kingdom, 7 School of Tropical Medicine and Global Health, Nagasaki University, Nagasaki, Japan

* rosamia.arao@gmail.com

**Data Availability Statement:** Survey dataset contains privacy-sensitive information including participant's individual and household income that

## Abstract

### Background

Tuberculosis (TB) is a disease associated with poverty. Moreover, a significant proportion of TB patients face a substantial financial burden before and during TB care. One of the top targets in the End TB strategy was to achieve zero catastrophic costs due to TB by 2020. To assess patient costs related to TB care and the proportion of TB-affected households that faced catastrophic costs, the Philippines National TB Programme (NTP) conducted a national TB patient cost survey in 2016–2017.

### Methods

A cross-sectional survey of 1,912 TB patients taking treatment in health facilities engaged with the NTP. The sample consists of 786 drug-sensitive TB (DS-TB) patients in urban facilities, 806 DS-TB patients in rural facilities, and 320 drug-resistant TB (DR-TB) patients. Catastrophic cost due to TB is defined as total costs, consisting of direct medical and non-medical costs and indirect costs net of social assistance, exceeding 20% of annual household income.

### Results

The overall mean total cost including pre- and post-diagnostic costs was US$601. The mean total cost was five times higher among DR-TB patients than DS-TB patients. Direct non-medical costs and income loss accounted for 42.7% and 40.4% of the total cost of TB, respectively. More than 40% of households had to rely on dissaving, taking loans, or selling their assets to cope with the costs. Overall, 42.4% (95% confidence interval (95% CI): 40.2–

formed a core part of the analysis. Even though we remove patient's identifiers such as patient number and name, there is still a possibility that those who are familiar with the project sites and beneficiaries may be able to identify participants and their households. The informed consent signed by all participants explicitly mentioned that only the research team have access to the data set. Due to such ethical and confidentiality restrictions, the survey dataset will be made available only upon request and with permission from the National Tuberculosis Control Programme (NTP), Department of Health, Philippines. All interested researchers will contact - the National TB Programme of the Philippines (ntp.mne@gmail.com), and - Non-author contact: Jose Gerard B. Belimac, MD, MPH (tuberculosis@doh.gov.ph) Team Lead, Infectious Diseases and Adult Health Division (Concurrent) and, Evidence Generation and Management Division DOH San Lazaro Compound, Rizal Ave. Sta. Cruz, Manila 1003 to request the data access.

**Funding:** The national TB patient cost survey in Philippines was financially supported by the Government of the United States of America through the United States Agency for International Development (USAID) and the Global Fund Grant for TB in the Philippines with Philippine Business for Social Progress as its principal recipient. The funders had no role in study design, data collection and analysis, decision to publish, or preparation of the manuscript. This article's contents are the responsibility of the authors and do not necessarily reflect the views of the sponsors.

**Competing interests:** The authors have declared that no competing interests exist.

44.6) of TB-affected households faced catastrophic costs due to TB, and it was significantly higher among DR-TB patients (89.7%, 95%CI: 86.3–93.0). A TB enabler package, which 70% of DR-TB patients received, reduced catastrophic costs by 13.1 percentage points (89.7% to 76.6%) among DR-TB patients, but only by 0.4 percentage points (42.4% to 42.0%), overall.

## Conclusions

TB patients in the Philippines face a substantial financial burden due to TB despite free TB services provided by the National TB Programme. The TB enabler package mitigated catastrophic costs to some extent, but only for DR-TB patients. Enhancing the current social and welfare support through multisectoral collaboration is urgently required to achieve zero catastrophic costs due to TB.

## Introduction

Tuberculosis (TB) is a cause and consequence of poverty [1]. Even though public health facilities provide free health care services for TB, patients frequently incur a heavy financial burden for seeking diagnosis, treatment, and care for TB [2, 3]. Such costs can be a significant barrier to accessing health care services and adhering to TB treatment, resulting in poor treatment outcomes and may increase further community transmission of TB as consequences [4–6].

Previous systematic reviews revealed that TB patients often face a heavy financial burden from TB, not only attributable to the medical expenses during TB treatment but also before TB treatment (while seeking a TB diagnosis) and income loss before and during TB treatment [7–9]. In one systematic review, income loss accounted for the largest proportion of the total costs incurred by TB patients at 60%, and this loss dominated the costs both before and during TB treatment [7]. Total TB-related costs incurred by TB-affected households as a proportion of their annual household income was 39%, and the proportions were higher among lower-income groups (89%) and patients with DR-TB (76–223%) [7, 10–13].

Given the economic impact of TB on TB-affected households as well as the global disease burden of TB, the World Health Organization (WHO) set the End TB Strategy, with one of its targets directed at ensuring "no family is burdened with catastrophic expenses due to TB" [14]. To capture the current situation of TB-associated household costs and monitor the progress to achieve this goal, WHO recommends countries with a high TB burden to conduct baseline and periodic TB patient cost surveys [15]. To implement the survey, WHO developed a generic protocol and data collection tool designed to assess various types of TB-related costs (e.g., direct medical and non-medical costs and indirect costs). "Catastrophic costs due to TB" refers to total TB-related costs exceeding 20% of the annual income of TB-affected households before having TB [15].

The Philippines is one of the top 30 high TB burden countries with an estimated TB incidence of 554 per 100,000 as of 2019 (3rd highest country globally) [16]. However, the financial burden of TB in affected households was not well known in the Philippines. Therefore, this study aimed to assess the magnitude and main drivers of patient costs to guide policies on cost mitigation. It also provides information on the risk factors that push TB patients to incur catastrophic costs during the course of treatment and the different cost drivers faced by drug-sensitive TB (DS-TB) and drug-resistant TB (DR-TB) patients.

## Methods

### Study setting

The Philippines is a country in the Western Pacific Region, comprised of more than 7,500 islands with a population of around 101 million in 2015 [17]. The country is a lower-middle-income country with a gross national income per capita of US$3,850 in 2019 [18]. The country has 17 Regions, 117 provinces and highly urbanized cities. Rural Health Units (RHUs) and urban health centers serve as TB Basic Management Units (BMUs). The most recent National TB Prevalence Survey in 2016 showed that the TB prevalence rate in the adult population was 1,159 per 100,000, which was 2.5 times higher than what was previously estimated for that year [19].

A national health insurance scheme has been implemented by the Philippines Health Insurance Corporation (PhilHealth) from 1995, and the population coverage of PhilHealth reached over 90% in 2019. Health insurance for TB covers costs for TB diagnosis, treatment, reporting, drugs, consultation, and health education [20, 21]. However, health facilities received reimbursement claims from PhilHealth for only about 11% of the notified TB patients in 2016 [22]. The Department of Social Welfare and Development administers a conditional cash transfer program for the poorest families with school-age children [23].

In the Philippines, a TB enabler package is provided by the NTP for DR-TB patients only, using a Global Fund grant. The package provides transportation, food, and accommodation for DR-TB patients to improve their adherence to TB treatment [24].

### Study design

A cross-sectional study was conducted between November 2016 and August 2017 to determine the magnitude of costs faced by patients seeking treatment in the NTP network. All TB patients (DS-TB and DR-TB cases) currently on treatment in an NTP facility who had completed at least two weeks of treatment in their current treatment phase (either intensive or continuation phase) were eligible to participate in the survey.

### Sample size

The total sample size of the survey was estimated at 1,880 TB patients subdivided into three subdomains (urban DS-TB, rural DS-TB, and DR-TB) to consider cost differences by drug resistance status (DS-TB vs. DR-TB cases) and by place of residence among DS-TB patients (urban vs. rural). For DS-TB patients, the required sample size was estimated at 780 for each subdomain (1,560 for urban and rural subdomains) using the following assumptions: a design effect of 1.81, an estimated catastrophic cost prevalence of 20%, a relative precision of 20%, and participation rate of 90%. For DR-TB patients, using an estimated catastrophic cost prevalence of 50% and the same assumptions for the other parameters used for DS-TB patients, the sample size was calculated to be 320 DR-TB patients.

A cluster sampling method was applied to ensure the national representativeness of the survey sample. The primary sampling unit was the Basic Management Unit (BMU) of the NTP. Using probability proportional to size (PPS), a total of 188 BMUs (78 for urban DS-TB, 78 for rural DS-TB, and 32 for DR-TB) were randomly selected based on TB case notification in 2015 for each subdomain. For each cluster, ten TB patients were randomly selected. In cases where the selected facility had less than ten eligible patients, a spare facility was used to augment the sample.

### Data collection

A locally adapted version of the World Health Organization (WHO) protocol and patient cost survey instrument was used for the survey [15]. The survey instrument was translated into

Tagalog, Cebuano, Ilocano, and Ilonggo which are commonly spoken native languages in the Philippines. The patients were randomly selected within each BMU using the Integrated Tuberculosis Information System (ITIS) of the NTP. Most patients interviewed were 18 years old or over. For patients 18 years old and below, parents, guardians, or any adult in the family knowledgeable of the family finances responded.

Data collection was conducted by a professional survey firm, Kantar-TNS. Prior to field deployment, interviewers received training on the survey instrument, the NTP program, and TB infection control measures. Showcards with illustrations and guides on diagnostic and laboratory tests were used to ensure a standardized interview process. Interviews were conducted at the health facility or at home, depending on the patient's availability and preference. The in-person interview lasted at most one hour and thirty minutes.

All survey forms underwent double encoding. The study team conducted quality and consistency checks in consultation with WHO. Consultations with NTP managers and health workers also helped verify the reported average expenses on various expense items and the frequency of facility visits, meeting with treatment partners, etc.

## Measurement of patient costs

Patient costs assessed in this survey consisted of three main cost components which are direct medical, direct non-medical, and indirect costs. Direct costs included out-of-pocket expenditures for TB care such as medical expenses for diagnosis and treatment (hospitalization, consultation fees, radiography, laboratory test, other procedures, medicines, and prescribed nutritional and food supplements) and the associated non-medical costs (transportation, food, and accommodation costs for patient and household members during facility visits, and additional non-prescribed supplements or foods in households). Indirect costs refer to the income loss incurred by TB patients and their households due to TB health care seeking and treatment. We estimated income loss in TB patients' households using the income before the current TB episode and that at the time of interview (output approach).

Pre-diagnostic costs refer to all medical and non-medical costs incurred from the onset of symptoms to the diagnosis of TB net of any reimbursement or social supports. From TB patients who were in the intensive phase, retrospective data on costs and time spent for care seeking before TB diagnosis were collected to estimate pre-diagnosis costs, however these questions were not asked of patients in the continuation phase to avoid recall bias.

Respondents were interviewed only once and reported on out-of-pocket (OOP) expenses related to TB diagnosis and treatment as well as household incomes, retrospectively. Since patients were interviewed at different stages of their treatment, some in the intensive treatment phase and others in the continuation treatment phase, only expenses incurred during the current treatment phase were asked. Moreover, to reduce recall bias, respondents were asked to report only the expenses incurred during the last visit to the facility for directly observed therapy (DOT), pick-up drugs, and medical follow-up visits. Finally, the respondents were also asked about the frequency of visits per month or week.

Information on costs incurred at the last visit and the frequency of visits was used to estimate total monthly expenses. To avoid overestimating costs, travel and food expenses for drug pick-up were not included for patients who reported going to the facility for DOT-related visits. Estimation of expense for the entire treatment phase used the following information: (i) total planned treatment duration (e.g., six months for DS-TB and 12 months for DR-TB); (ii) expenses reported by the patient; and (iii) information coming from other respondents, particularly the median (**Table 1**) according to the WHO methodology [15]. For example, for a patient interviewed at the intensive phase, the total expense for that phase is estimated by

**Table 1. Type of patient, treatment phase during interview, and collected information.**

| Type of patient | Before treatment | During treatment | |
|---|---|---|---|
| | | **Intensive** | **Continuation** |
| New, interviewed at the intensive phase | Medical | Hospitalization | **NOT COLLECTED. ESTIMATED** |
| | Non-medical | DOT visit | |
| | | Picking up of drugs | |
| | | Follow-up visit | |
| | | Nutritional and food supplement | |
| | | Enablers and other support | |
| | | Income loss | |
| New, interviewed at the continuation phase | **NOT COLLECTED. ESTIMATED** | **NOT COLLECTED. ESTIMATED** | Hospitalization |
| | | | DOT visit |
| | | | Picking up of drugs |
| | | | Follow-up visit |
| | | | Nutritional and food supplement |
| | | | Enablers and other support |
| | | | Income loss |
| Retreatment, interviewed at the intensive phase | **NOT COLLECTED. ESTIMATED** | Hospitalization | **NOT COLLECTED. ESTIMATED** |
| | | DOT visit | |
| | | Picking up of drugs | |
| | | Follow-up visit | |
| | | Nutritional and food supplement | |
| | | Enablers and other support | |
| | | Income loss | |
| Retreatment, interviewed at the continuation phase | **NOT COLLECTED. ESTIMATED** | **NOT COLLECTED. ESTIMATED** | Hospitalization |
| | | | DOT visit |
| | | | Picking up of drugs |
| | | | Follow-up visit |
| | | | Nutritional and food supplement |
| | | | Enablers and other support |
| | | | Income loss |

* pre-treatment income had been collected for all patients.

multiplying reported expense per visit with the frequency of visit and treatment duration for this phase. To estimate the possible expense for the continuation phase, TB patients' median expense at the continuation phase in the same subdomain is used. This approach simplified sampling and data collection since no follow-ups were required for respondents. Table 1 shows the information collected by type of patient and treatment phase at the time of the interview.

Before the interview, patients were advised to bring along a family member familiar with the household's monthly earnings. Indirect cost was estimated as income loss, which is the difference between the family income of all members before and during the TB illness. Income loss related to the TB treatment was estimated using the planned treatment duration. For DR-TB patients with a treatment duration of more than twelve months, income loss was estimated for only up to twelve months, following the WHO recommended method. If monthly income during TB treatment was higher than the monthly income before illness, income loss was estimated to be zero.

## Data analysis

Data cleaning and processing were carried out using Stata 16.0 (StataCorp 2020). Statistical analyses and data visualizations were performed using R4.0.2 (CRAN: Comprehensive R Archive Network at https://cran.r-project.org/). Mean with standard deviation (SD) and median with inter-quartile range (IQR) were used for continuous data, and frequency was presented for categorical data. Statistical differences between urban and rural DS-TB patients and DR-TB patients were examined using a chi-square test for categorical data and the ANOVA or Kruskal–Wallis test for continuous data. Statistical significance was defined as a p-value less than 0.05. Information on costs and incomes were collected in the Philippines Pesos (Php) and later converted into US$ for analysis at the rate of Php 50.11 per US$ 1 as of March 31, 2017 (Oanda.com). Clustering effects related to the sampling method and sampling probabilities were adjusted for in estimating overall patient costs and proportion of TB-affected households facing catastrophic costs.

Catastrophic cost due to TB was defined as total costs, consisting of direct medical and non-medical costs and indirect costs, exceeding 20% of annual household income. This definition is consistent with the global End TB indicator defined by the WHO. Additionally, a sensitivity analysis was performed to investigate how varying the threshold (to different percentages from 10% to 30%) affected the proportion of catastrophic costs. Furthermore, we carried out univariate logistic regression analysis to identify demographic, clinical, and economic factors associated with facing catastrophic costs. Then, we performed a stepwise analysis with forward selection to build the final multivariate model. The results of univariate and multivariate logistic regression analysis were presented using odds ratios (OR) and 95% confidence interval (95% CI).

## Ethical considerations

The St. Cabrini Medical Center-Asian Eye Institute Ethics Review Committee (SCMC-AEI ERC) reviewed and provided a national ethics approval for the study protocol and survey instrument (ERC #2016–020). Individual ethics approvals were also obtained from Baguio General Hospital and Medical Center Ethics Review Committee (BGH-MC ERC), Department of Health XI Cluster Ethics Review Committee (DOH XI CERC), and the National Children's Hospital Institutional Review Board (NCH-IRB) for the three cluster facilities with their own Internal Review Board (IRB).

A written consent form was obtained from all participants before the commencement of the interview. The informed consent signed by all participants explicitly mentioned that only the research team have access to the dataset. For participants less than 18 years old, we obtained written consent from their parents or legal guardians who accompanied them.

## Results

### Study population

A total of 1,912 TB patients were enrolled and interviewed in this survey. Of these, 786 were DS-TB patients treated in urban facilities (urban DS-TB), 806 were DS-TB patients treated in rural facilities (rural DS-TB), and 320 were DR-TB patients (**Table 2**). Overall, 65.0% of survey participants were males, and the mean age was 40.2. Children less than 15 years old consisted of 9.4% of survey participants. The proportion of participants with no education or only with pre- and elementary level education was 32.1%. More than half (58.9%) were enrolled in health insurance(s). About half (52.0%) of the survey participants were employed before TB diagnosis, and 41.7% were the breadwinner in their households. The mean monthly household income was US$246 before TB diagnosis, which declined to US$206 during TB treatment.

**Table 2. Socio-demographic characteristics of participants.**

| Characteristics | | Urban DS-TB | Rural DS-TB | DR-TB | Overall | p-value |
|---|---|---|---|---|---|---|
| | | N = 786 | N = 806 | N = 320 | N = 1,912 | |
| **Sex** | Female | 276 (35.1%) | 292 (36.2%) | 102 (31.9%) | 670 (35.0%) | 0.385 |
| | Male | 510 (64.9%) | 514 (63.8%) | 218 (68.1%) | 1242 (65.0%) | |
| **Age** | Mean (SD) | 37.6 (19.1) | 41.7 (21.6) | 42.9 (14.0) | 40.2 (19.6) | <0.001 |
| **Age group (years)** | 0–14 | 72 (9.2%) | 108 (13.4%) | 0 (0.0%) | 180 (9.4%) | <0.001 |
| | 15–24 | 172 (21.9%) | 85 (10.5%) | 37 (11.6%) | 294 (15.4%) | |
| | 25–34 | 124 (15.8%) | 113 (14.0%) | 60 (18.8%) | 297 (15.5%) | |
| | 35–44 | 124 (15.8%) | 125 (15.5%) | 76 (23.8%) | 325 (17.0%) | |
| | 45–54 | 111 (14.1%) | 117 (14.5%) | 77 (24.1%) | 305 (16.0%) | |
| | 55–64 | 110 (14.0%) | 131 (16.3%) | 47 (14.7%) | 288 (15.1%) | |
| | 65+ | 73 (9.3%) | 127 (15.8%) | 23 (7.2%) | 223 (11.7%) | |
| **Education level** | No education | 4 (0.5%) | 13 (1.6%) | 7 (2.2%) | 24 (1.3%) | <0.001 |
| | Pre-/elementary level | 182 (23.2%) | 321 (39.8%) | 86 (26.9%) | 589 (30.8%) | |
| | High-school/secondary level | 421 (53.6%) | 323 (40.1%) | 155 (48.4%) | 899 (47.0%) | |
| | College level | 175 (22.3%) | 143 (17.7%) | 70 (21.9%) | 388 (20.3%) | |
| | Unknown | 4 (0.5%) | 6 (0.7%) | 2 (0.6%) | 12 (0.6%) | |
| **Health insurance** | No | 393 (50.0%) | 267 (33.1%) | 126 (39.4%) | 786 (41.1%) | <0.001 |
| | Yes | 393 (50.0%) | 539 (66.9%) | 194 (60.6%) | 1126 (58.9%) | |
| **Employment status before TB** | Employed | 381 (48.5%) | 439 (54.5%) | 175 (54.7%) | 995 (52.0%) | 0.033 |
| | Unemployed | 405 (51.5%) | 367 (45.5%) | 145 (45.3%) | 917 (48.0%) | |
| **Patient was main income earner** | No | 470 (59.8%) | 477 (59.2%) | 168 (52.5%) | 1115 (58.3%) | 0.067 |
| | Yes | 316 (40.2%) | 329 (40.8%) | 152 (47.5%) | 797 (41.7%) | |
| **Household size** | 1–6 | 315 (40.1%) | 400 (49.6%) | 167 (52.2%) | 882 (46.1%) | <0.001 |
| | 7+ | 471 (59.9%) | 406 (50.4%) | 153 (47.8%) | 1030 (53.9%) | |
| **Monthly household income (pre-diagnosis)** | Mean (SD), US$ | 282 (304) | 210 (322) | 251 (330) | 246 (318) | <0.001 |
| **Monthly household income (during TB)** | Mean (SD), US$ | 243 (271) | 185 (319) | 170 (243) | 206 (290) | <0.001 |

SD: standard deviation, DS-TB: drug-susceptible TB, DR-TB: drug-resistant TB.

* Overall statistics is summary for our sample, not weighted (not adjusted for sampling probabilities).

The mean age of urban DS-TB patients (37.6 years) was significantly lower than rural DS-TB and DR-TB (Rural DS-TB: 41.7 years, DR-TB: 42.9 years, p<0.001), and the proportion with higher education (high school, secondary level, and college level) was higher among urban DS-TB patients (urban DS-TB: 75.9%, rural DS-TB: 57.8%, DR-TB: 70.3%, p<0.001). The mean of monthly household income was significantly higher before having TB among urban DS-TB patients compared to that among rural DS-TB and DR-TB patients (urban DS-TB: US$282, rural DS-TB: US$210, DR-TB: US$251, p<0.001), and the same tendency was also observed in monthly household income during TB treatment. The reduction in mean monthly household income from pre-diagnosis to post-diagnosis was observed among all three categories of TB patients.

Newly diagnosed TB was predominant among DS-TB patients (85.4% in urban DS-TB and 90.2% in rural DS-TB), while it was only 10.3% in DR-TB patients. The proportion of extra-pulmonary TB (EPTB) was 1.5% in urban DS-TB patients and 1.2% in rural DS-TB patients, respectively (**Table 3**). Most of the participants were taking TB treatment with treatment partner(s) (84.5%). Forty-eight percent reported a delay in TB diagnosis (defined as a duration between onset of TB symptoms and TB diagnosis of four weeks or more), and the median duration from the onset of TB symptoms until having TB diagnosis was 3.9 weeks. A high

Table 3. Clinical characteristics of participants.

| Characteristics | | Urban DS-TB | Rural DS-TB | DR-TB | Overall | p-value |
|---|---|---|---|---|---|---|
| | | N = 786 | N = 806 | N = 320 | N = 1,912 | |
| Registration group | New | 671 (85.4%) | 727 (90.2%) | 33 (10.3%) | 1431 (74.8%) | <0.001 |
| | Relapse | 85 (10.8%) | 57 (7.1%) | 169 (52.8%) | 311 (16.3%) | |
| | Retreatment | 28 (3.6%) | 18 (2.2%) | 117 (36.6%) | 163 (8.5%) | |
| | Unknown | 2 (0.3%) | 4 (0.5%) | 1 (0.3%) | 7 (0.4%) | |
| Drug resistance status | DS-TB | 786 (100.0%) | 806 (100.0%) | 0 (0.0%) | 1592 (83.3%) | <0.001 |
| | DR-TB | 0 (0.0%) | 0 (0.0%) | 320 (100.0%) | 320 (16.7%) | |
| Site of disease | Pulmonary TB | 774 (98.5%) | 796 (98.8%) | 320 (100.0%) | 1890 (98.8%) | 0.093 |
| | Extra-pulmonary TB | 12 (1.5%) | 10 (1.2%) | 0 (0.0%) | 22 (1.2%) | |
| Mode of diagnosis | Bacteriologically confirmed | 326 (41.5%) | 364 (45.2%) | 320 (100.0%) | 1010 (52.8%) | <0.001 |
| | Clinically diagnosed | 460 (58.5%) | 442 (54.8%) | 0 (0.0%) | 902 (47.2%) | |
| Treatment phase | Intensive phase | 155 (19.7%) | 98 (12.2%) | 118 (36.9%) | 371 (19.4%) | <0.001 |
| | Continuation phase | 631 (80.3%) | 708 (87.8%) | 202 (63.1%) | 1541 (80.6%) | |
| Mode of treatment | Self-administered | 162 (20.6%) | 115 (14.3%) | 12 (3.8%) | 289 (15.1%) | <0.001 |
| | With Treatment partner | 624 (79.4%) | 684 (84.9%) | 308 (96.2%) | 1616 (84.5%) | |
| | Other | 0 (0.0%) | 7 (0.9%) | 0 (0.0%) | 7 (0.4%) | |
| Delay before diagnosis (>4weeks) | No | 97 (56.1%) | 33 (47.1%) | 15 (40.5%) | 145 (51.8%) | 0.153 |
| | Yes | 76 (43.9%) | 37 (52.9%) | 22 (59.5%) | 135 (48.2%) | |
| Weeks before TB diagnosis | Median (IQR) | 3.4 (1.4–8.4) | 4.2 (1.1–12.4) | 5.9 (0.0–42.7) | 3.9 (1.3–10.3) | 0.136 |
| Hospitalized at the time of interview | No | 779 (99.1%) | 789 (97.9%) | 307 (95.9%) | 1875 (98.1%) | 0.002 |
| | Yes | 7 (0.9%) | 17 (2.1%) | 13 (4.1%) | 37 (1.9%) | |
| Hospitalized during current phase until the time of interview | No | 767 (97.6%) | 777 (96.4%) | 298 (93.1%) | 1842 (96.3%) | 0.002 |
| | Yes | 19 (2.4%) | 29 (3.6%) | 22 (6.9%) | 70 (3.7%) | |
| Number of facility visits, mean (SD) | Pre-diagnosis | 1.2 (0.9) | 1.1 (0.8) | 1.1 (0.6) | 1.1 (0.8) | 0.870 |
| | Directly observed therapy | 155.7 (37.4) | 162.4 (35.4) | 475.1 (90.3) | 219.4 (134.2) | <0.001 |
| | Drug pick-up | 134.4 (31.4) | 112.5 (53.3) | 408.0 (112.3) | 167.7 (118.8) | <0.001 |
| | Follow-up | 7.5 (3.7) | 8.1 (3.5) | 24.3 (11.4) | 10.6 (8.4) | <0.001 |

SD: standard deviation, DS-TB: drug-susceptible TB, DR-TB: drug-resistant TB.

* Overall statistics is a summary of our sample, not weighted (not adjusted for sampling probabilities).

frequency of facility visits was observed for directly observed therapy (DOT) (219.4 times) and drug pick-up (118.8 times).

The proportion of relapse or retreatment cases was higher in DR-TB patients than DS-TB patients (DR-TB: 89.4%, urban DS-TB: 14.4%, rural DS-TB: 9.3%, p<0.001). All DR-TB patients were diagnosed with bacteriological confirmation, whereas more than half of DS-TB patients were with clinical diagnosis (DR-TB: 0%, urban DS-TB: 58.5%, rural DS-TB: 54.8%, p<0.001). DR-TB patients experienced more facility visits for DOT, drug pick-up, and medical follow-up than DS-TB patients due to the longer treatment duration (DOT visits: DR-TB 475.1 times, urban DS-TB 155.7 times, rural DS-TB 162.4 times, p<0.001; drug pick-up visits: DR-TB 408.0 times, urban DS-TB 134.4 times, rural DS-TB 112.5 times, p<0.001; medical follow-up visits: DR-TB 24.3 times, urban DS-TB 7.5 times, rural DS-TB 8.1 times, p<0.001).

## Time loss and costs incurred by TB patients and their households

Mean total lost time for all TB patients and their caregivers was 157.4 hours and 54.7 hours, respectively, with a significantly longer lost time for DR-TB patients than DS-TB patients (patients: DR-TB 600.9 hours, urban DS-TB 59.5 hours, rural DS-TB 76.9 hours, p<0.001;

**Table 4. Time loss for TB care-seeking.**

| Time loss due to TB (working hour basis) | | Urban DS-TB | Rural DS-TB | DR-TB | Overall | p-value |
|---|---|---|---|---|---|---|
| | | N = 786 | N = 806 | N = 320 | N = 1,912 | |
| **Hours lost by patient, mean (SD)** | Pre-disease | 18.1 (42.3) | 16.1 (24.0) | 31.4 (65.6) | 18.2 (38.7) | 0.590 |
| | Hospitalization | 86.3 (82.1) | 104.6 (123.9) | 383.1 (723.2) | 189.1 (434.3) | 0.027 |
| | Directly observed therapy | 41.9 (70.5) | 53.6 (117.6) | 497.8 (627.4) | 123.1 (319.1) | <0.001 |
| | Drug pick-up | 3.7 (18.0) | 3.8 (33.7) | 28.0 (226.3) | 7.8 (96.1) | <0.001 |
| | Medical follow-up | 6.3 (9.7) | 10.5 (14.3) | 40.0 (46.5) | 13.7 (25.1) | <0.001 |
| | Total lost time | 59.5 (77.8) | 76.9 (130.2) | 600.9 (721.9) | 157.4 (369.1) | <0.001 |
| **Hours lost by care giver, mean (SD)** | Hospitalization | 53.4 (95.9) | 61.5 (127.2) | 316.4 (726.2) | 149.8 (453.0) | 0.080 |
| | Directly observed therapy | 16.2 (51.8) | 34.2 (93.9) | 181.2 (414.2) | 59.4 (214.4) | <0.001 |
| | Drug pick-up | 1.2 (10.5) | 2.0 (15.1) | 15.9 (198.5) | 4.2 (86.2) | 0.028 |
| | Medical follow-up | 2.0 (7.6) | 5.2 (13.0) | 12.3 (35.3) | 5.2 (18.3) | <0.001 |
| | Total lost time | 14.1 (49.1) | 31.1 (90.8) | 213.8 (520.9) | 54.7 (234.3) | <0.001 |

SD: standard deviation, DS-TB: drug-susceptible TB, DR-TB: drug-resistant TB.

* Overall statistics is a summary of our sample, not weighted (not adjusted for sampling probabilities).

** Time loss for each category of facility visits was estimated from the patients who experienced each type of the visits.

*** Time loss which was reported as more than 1 day (e.g. for hospitalizations) was converted to working hour basis based on 8 hours per day.

caregivers: DR-TB 213.8 hours, urban DS-TB 14.1 hours, rural DS-TB 31.1 hours, p<0.001) (**Table 4**). Overall, substantially more time was lost for hospitalization (189.1 hours) and DOT (123.1 hours). This was mostly for DR-TB patients compared to DS-TB patients (hospitalization: DR-TB 383.1 hours, urban DS-TB 86.3 hours, rural DS-TB 104.6 hours, p = 0.027; DOT: DR-TB 497.8 hours, urban DS-TB 41.9 hours, rural DS-TB 53.6 hours, p<0.001).

The overall total mean cost was estimated at US$601.4, equivalent to 2.4 times the reported monthly household income (pre-diagnosis) (**Table 5**). The total cost was mainly driven by

**Table 5. Estimated mean total costs incurred by TB-affected households, assessed by output approach in US$.**

| TB patient costs, mean (SD), US$ | | | Urban DS-TB | Rural DS-TB | DR-TB | Overall (weighted) |
|---|---|---|---|---|---|---|
| | | | N = 786 | N = 806 | N = 320 | N = 1,912 |
| **Pre-TB diagnosis** | Direct medical costs | | 16.2 (16.9) | 16.4 (27.9) | 20 (208.6) | 16.4 (38.8) |
| | Direct non-medical costs | | 2.6 (4.9) | 3.1 (7.8) | 6.5 (76.8) | 3.1 (10.6) |
| | Total direct costs | | 18.8 (20.3) | 19.5 (33.7) | 26.4 (285) | 19.5 (47) |
| **Post-TB diagnosis** | Direct medical costs | Pick-up | 0.7 (20.5) | 0 (0) | 0 (0) | 0.1 (5.2) |
| | | Directly observed therapy | 0 (0) | 0 (0) | 0 (0) | 0 (0) |
| | | Follow-up | 45.8 (156.3) | 74.4 (271.7) | 135.8 (913.7) | 70.7 (353.5) |
| | | Hospitalization | 8.7 (90.6) | 14.7 (112.2) | 62.6 (730) | 14.5 (147) |
| | Direct non-medical costs | Travel | 35.8 (79.2) | 93 (370.2) | 774.2 (967.5) | 94.1 (476.5) |
| | | Accommodation | 7.2 (101.9) | 1.9 (33) | 13.1 (92.8) | 2.9 (49.3) |
| | | Food | 18.8 (63.1) | 44.1 (125.3) | 356.2 (469.7) | 44.8 (164.3) |
| | | Nutrition supplement | 113.3 (174.6) | 104.8 (136.6) | 472.6 (511.8) | 111.9 (196) |
| **Total direct medical costs** | | | 71.4 (187.2) | 105.5 (311.3) | 218.4 (1191.3) | 101.7 (409) |
| **Total direct non-medical costs** | | | 177.6 (253.9) | 246.9 (479.7) | 1622.6 (1346.6) | 256.8 (642.3) |
| **Indirect costs (income loss)** | | | 330.1 (987.4) | 210 (553.3) | 1078.5 (3000.7) | 242.9 (765.5) |
| **Total cost (output approach)** | | | 579.2 (1082.9) | 562.4 (835.3) | 2919.5 (3532.5) | 601.4 (1178.4) |

SD: standard deviation, DS-TB: drug-susceptible TB, DR-TB: drug-resistant TB.

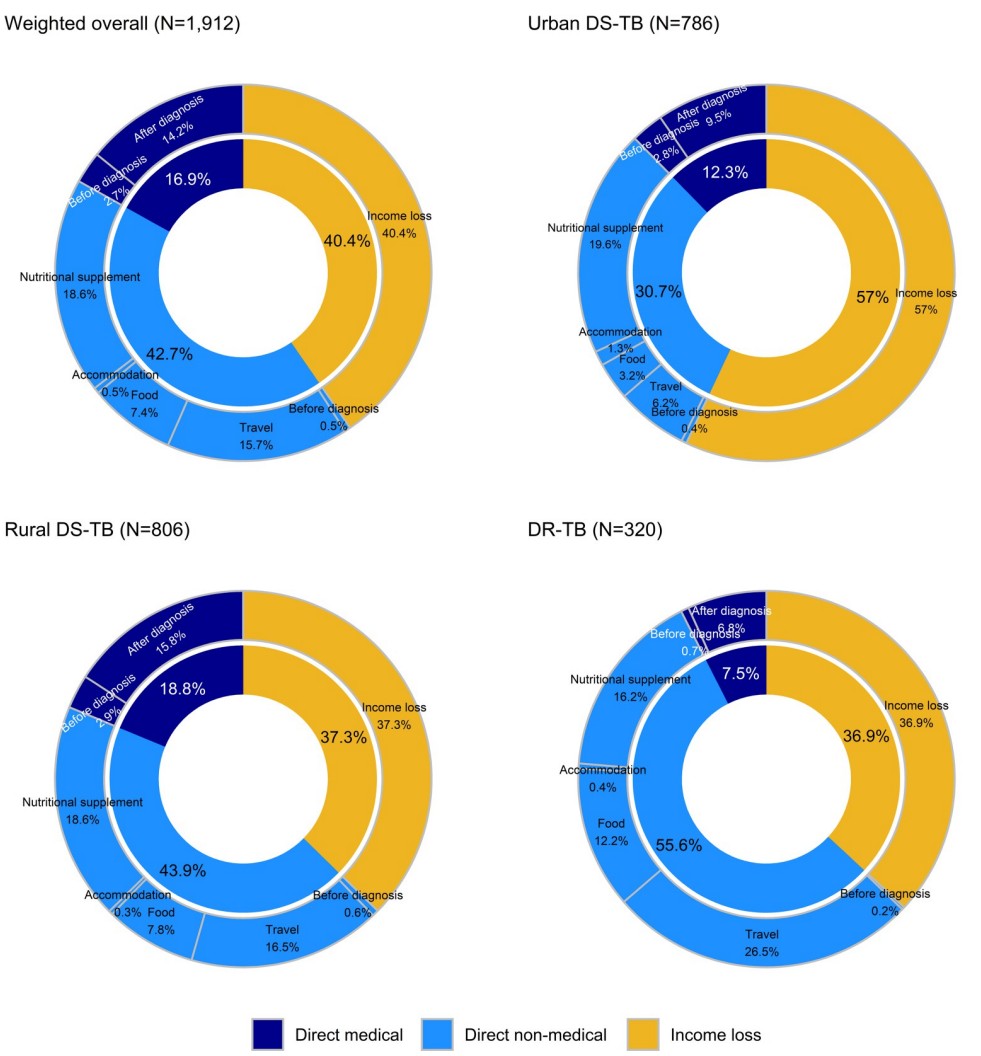

**Fig 1. Composition of TB patient costs by urban/rural and drug resistance status.**

direct non-medical costs (42.7%) followed by indirect costs (income loss) (40.4%) and direct medical costs (16.9%) (**Fig 1**). Particularly, travel costs for facility visits and costs for additional food and non-prescribed supplements other than regular diets comprised a large proportion among components of direct non-medical costs (travel costs: 15.7%, nutritional supplements: 18.6% of total costs). Direct medical costs were incurred mainly in post-TB-diagnosis (US$ 85.2, 14.2% of total costs), and costs incurred during medical follow-up visits were predominant (overall weighted mean US$70.7). Prescribed nutritional supplements other than TB medications accounted for a substantial share of direct medical costs during medical follow-up visits (45.9%).

The total mean cost was over five times greater among DR-TB patients compared to DS-TB patients (DR-TB: US$ 2,919.5, urban DS-TB: US$579.2, rural DS-TB: US$562.4). The main cost driver in rural DS-TB and DR-TB patients was direct non-medical costs (rural DS-TB: 43.9%, DR-TB: 55.6%), similar to the overall figure, whereas the total cost in urban DS-TB was largely driven by income loss (57.0%).

**Table 6. Reported coping mechanisms and social consequences.**

| Coping mechanism and social consequences | | Urban DS-TB | Rural DS-TB | DR-TB | Overall | p-value |
|---|---|---|---|---|---|---|
| | | N = 786 | N = 806 | N = 320 | N = 1,912 | |
| **Coping mechanism** | Dissaving | 100 (12.7%) | 113 (14.0%) | 37 (11.6%) | 250 (13.1%) | 0.506 |
| | Taking loans | 201 (25.6%) | 265 (32.9%) | 111 (34.7%) | 577 (30.2%) | 0.001 |
| | Selling household assets | 37 (4.7%) | 68 (8.4%) | 47 (14.7%) | 152 (7.9%) | <0.001 |
| | Any of above | 271 (34.5%) | 360 (44.7%) | 142 (44.4%) | 773 (40.4%) | <0.001 |
| **Unemployed** | Before TB diagnosis | 405 (51.5%) | 367 (45.5%) | 145 (45.3%) | 917 (48.0%) | 0.033 |
| | At the interview | 505 (64.2%) | 515 (63.9%) | 256 (80.0%) | 1276 (66.7%) | <0.001 |
| **Proportion below poverty line** | Before TB diagnosis | 491 (62.5%) | 595 (73.8%) | 211 (65.9%) | 1297 (67.8%) | <0.001 |
| | At the interview | 562 (71.5%) | 645 (80.0%) | 282 (88.1%) | 1489 (77.9%) | <0.001 |
| **Received social support/welfare** | Vouchers/enablers | 7 (0.9%) | 12 (1.5%) | 222 (69.4%) | 241 (12.6%) | <0.001 |
| | Conditional cash transfer | 3 (0.4%) | 10 (1.2%) | 11 (3.4%) | 24 (1.3%) | <0.001 |
| **Social consequences** | Interrupted schooling | 26 (3.3%) | 26 (3.2%) | 28 (8.8%) | 80 (4.2%) | <0.001 |
| | Divorce | 6 (0.8%) | 2 (0.2%) | 6 (1.9%) | 14 (0.7%) | 0.015 |
| | Social exclusion | 74 (9.4%) | 107 (13.3%) | 63 (19.7%) | 244 (12.8%) | <0.001 |
| | Food insecurity | 128 (16.3%) | 210 (26.1%) | 68 (21.2%) | 406 (21.2%) | <0.001 |
| | Job loss | 191 (24.3%) | 222 (27.5%) | 144 (45.0%) | 557 (29.1%) | <0.001 |
| | Any of above | 337 (42.9%) | 417 (51.7%) | 209 (65.3%) | 963 (50.4%) | <0.001 |
| **Perceived impact** | No impact | 230 (29.3%) | 194 (24.1%) | 44 (13.8%) | 468 (24.5%) | <0.001 |
| | Little impact | 178 (22.6%) | 175 (21.7%) | 47 (14.7%) | 400 (20.9%) | |
| | Moderate impact | 249 (31.7%) | 243 (30.1%) | 90 (28.1%) | 582 (30.4%) | |
| | Serious impact | 81 (10.3%) | 89 (11.0%) | 66 (20.6%) | 236 (12.3%) | |
| | Very serious impact | 48 (6.1%) | 105 (13.0%) | 73 (22.8%) | 226 (11.8%) | |

DS-TB: drug-susceptible TB, DR-TB: drug-resistant TB.

* Overall statistics is a summary of our sample, not weighted (not adjusted for sampling probabilities).

Overall, 40.4% of survey participants mobilized their savings, took loans, or sold their household assets to cope with the financial impacts due to TB care, and the proportion was higher among rural DS-TB and DR-TB patients than urban DS-TB patients (rural DS-TB: 44.7%, DR-TB: 44.4%, urban DS-TB patients: 34.5%, p<0.001) (**Table 6**). The proportion of patients who received vouchers as social supports was largely only found among DR-TB patients (69.4%). The proportion of households receiving conditional cash transfers for the poor was only 1.3%.

The proportion of households living below the poverty line (living with less than 1.9US$ per day) increased from 67.8% (pre-diagnosis) to 77.9% (at the time of interview) during TB treatment, and the impoverishment occurred particularly in DR-TB patients from 65.9% (pre-diagnosis) to 88.1% (at the time of interview). The unemployment rate also increased from 48.0% to 66.7%, and in DR-TB patients, the rate almost doubled from 45.3% to 80.0%. 29.1% experienced job loss, 21.2% faced food insecurity, and 12.8% experienced social exclusion due to TB. Among DR-TB patients, the proportion of households who had to interrupt their children's schooling (8.8%, p<0.001), and experiences of social exclusion (19.7%, p<0.001) and job loss (45.0%, p<0.001) was significantly higher while the proportion of facing food insecurity was higher in rural DS-TB patients (26.1%, p<0.001).

More than half of households (54.5%) reported that having TB had a moderate, severe, or very-severe economic impact in their households, and the proportion was significantly higher among DR-TB patients (DR-TB: 71.5%, urban DS-TB: 48.1%, rural DS-TB: 54.1%, p<0.001).

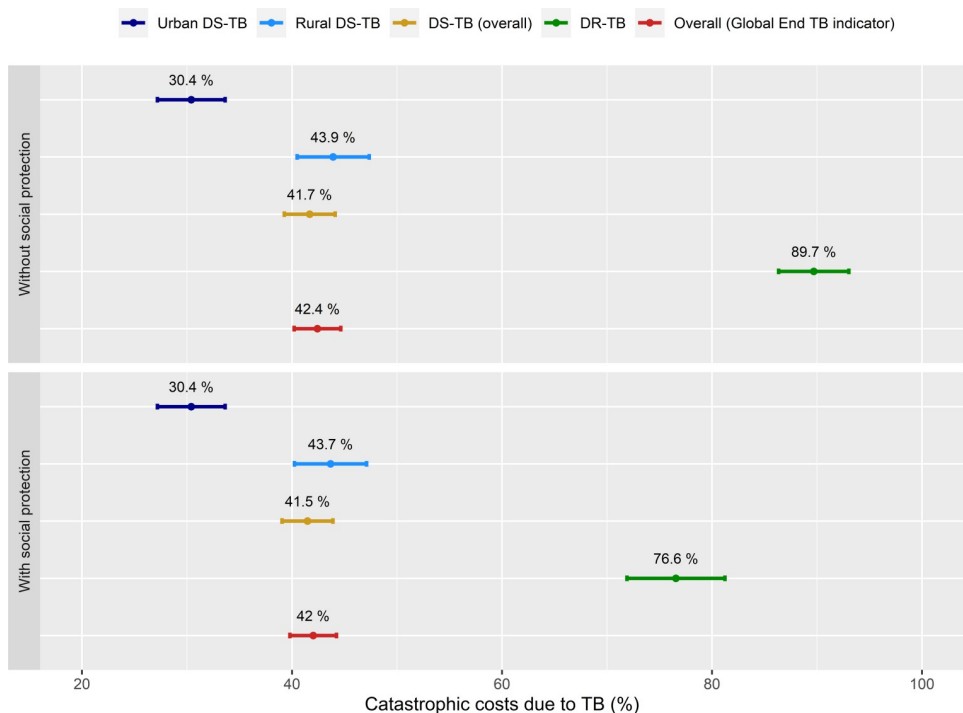

**Fig 2. Proportion of TB-affected households facing catastrophic costs using 20% threshold of annual household income.** * Overall statistics is adjusted for sampling probabilities.

## Proportion of households facing catastrophic costs and its risk factors

The proportion of TB-affected households experiencing catastrophic costs was 42.4% (95%CI: 40.2%-44.6%) using a 20% threshold of annual household income (**Fig 2**). This indicated that the proportion of TB patients would have experienced catastrophic costs if there were no social protection. This proportion was substantially higher in DR-TB patients (89.7%, 95% confidence interval or CI: 86.3%-93.0%) compared to that in DS-TB patients (weighted) (41.7%, 95%CI: 39.3%-44.1%). When considering the vouchers received by patients as compensation for patient expenses, the proportion of catastrophic costs in DR-TB patients was reduced from 89.7% to 76.6%, while it had only a minimal impact on the overall proportion of catastrophic costs (declined from 42.4% to 42.0%). The overall proportion of catastrophic costs ranged from 60.7% to 32.8% when the annual household income threshold was changed from 10% to 30% (**Fig 3**).

With adjustment for potential covariates in multivariate analysis, various factors were identified as risk factors associated with catastrophic costs such as living in rural areas (OR: 1.37, p = 0.005), being poorer (OR: 3.85, p<0.001 in poorest quintile), small household size (OR: 1.25, p = 0.044), with employment before TB (OR: 2.26, p<0.001), having DR-TB (OR:8.27, p<0.001), with previous treatment history (relapse OR: 2.14, p<0.001. retreatment OR: 2.08, p = 0.004), having TB treatment from treatment partners (OR: 1.62, p = 0.002), and being hospitalized during the current treatment phase (OR: 9.47, p<0.001) (**Table 7**).

## Discussion

This survey found that the overall proportion of catastrophic costs in TB-affected households was 42.4%, with the mean total costs of US$ 601.4, which was equivalent to 2.4 times the

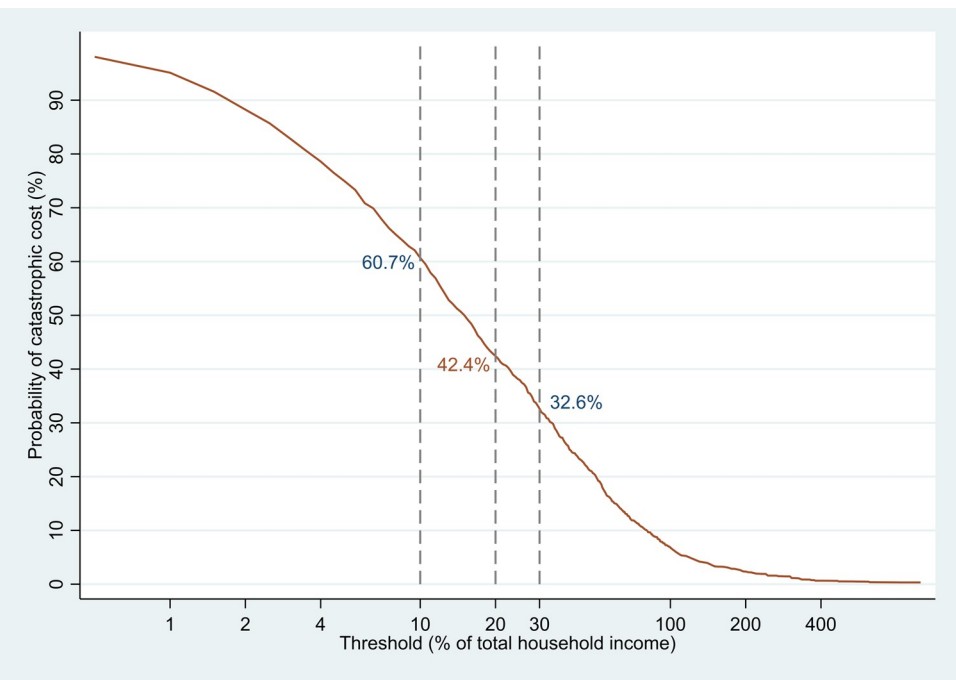

**Fig 3. Sensitivity analysis to define the proportion of TB-affected household experiencing catastrophic costs at different thresholds.** * The statistics is adjusted for sampling probabilities.

reported monthly household income of survey participants. The occurrence of catastrophic costs due to TB was significantly higher among DR-TB patients (89.7%) than DS-TB patients (41.7%). The overall main cost driver was direct non-medical costs (42.7%) followed by income loss (40.4%). Of the survey participants, 40.4% mobilized their savings, borrowed money, or sold their household assets to cope with the economic impacts of TB.

Implementation of national TB patient cost surveys started in Myanmar in 2015. As of September 2021, 25 countries had completed a survey, and 9 of them are countries in the Western Pacific Region, including China (2016), Fiji (2017), Lao PDR (2019), Mongolia (2017), Papua New Guinea (2019), Solomon Islands (2020), and Viet Nam (2016) [13, 25, 26]. The results showed that the proportion of catastrophic costs was varied depending on country settings, ranging from 19% in Lesotho to 92% in Solomon Islands [16, 27]. The overall pooled average of catastrophic costs using available results from 17 countries was 49% (95%CI: 34%-63%), and the disaggregated pooled average for DR-TB patients was 80% (95% CI: 70–90%) which was double that for DS-TB patients 44% (95% CI: 31–58%) [28]. Results from our survey in the Philippines were in the range of the global pooled average.

The total TB patient costs in the Philippines were largely driven by direct non-medical costs such as expenses for transportation and nutritional supplements and foods, and regular diet. Similar results were observed in 5 out of 25 countries such as Ghana, Kenya, Uganda, Lao PDR, and Timor Leste [28]. Malnutrition is a well-known risk factor associated with TB. Also, weight loss is a common clinical symptom of TB [29–31]. Therefore, TB patients usually require an increased intake of foods and nutrients (i.e., macronutrients) to recover from TB-associated weight loss and malnutrition. Hence, they may have to purchase and consume additional food apart from their regular diets. Although the survey did not collect data on the body mass index (BMI) of participants, a recent study in the Philippines observed that 36.6% of TB patients were moderately or severely malnourished (BMI<18.5) [32]. In other TB patient cost

**Table 7. Risk factors associated with facing catastrophic costs due to TB.**

| Risk factors | | Total | with catastrophic costs | Univariate analysis Crude OR (95%CI, p-value) | Multivariate analysis Adjusted OR (95%CI, p-value) |
|---|---|---|---|---|---|
| | | N | n (%) | | |
| Age group (years) | 25–34 | 297 | 153 (51.5%) | Ref | - |
| | 0–14 | 180 | 55 (30.6%) | 0.41 (0.28–0.61, p<0.001) | - |
| | 15–24 | 294 | 108 (36.7%) | 0.55 (0.39–0.76, p<0.001) | - |
| | 35–44 | 325 | 164 (50.5%) | 0.96 (0.70–1.31, p = 0.793) | - |
| | 45–54 | 305 | 156 (51.1%) | 0.99 (0.72–1.36, p = 0.928) | - |
| | 55–64 | 288 | 137 (47.6%) | 0.85 (0.62–1.18, p = 0.340) | - |
| | 65+ | 223 | 107 (48.0%) | 0.87 (0.61–1.23, p = 0.425) | - |
| Sex | Female | 670 | 299 (44.6%) | Ref | - |
| | Male | 1,242 | 581 (46.8%) | 1.09 (0.90–1.32, p = 0.368) | - |
| Urban/rural | Urban | 846 | 291 (34.4%) | Ref | Ref |
| | Rural | 1,066 | 589 (55.3%) | 2.36 (1.96–2.84, p<0.001) | 1.37 (1.10–1.71, p = 0.005) |
| Income quintile | Wealthiest | 382 | 136 (35.6%) | Ref | Ref |
| | Fourth | 358 | 129 (36.0%) | 1.02 (0.75–1.38, p = 0.903) | 1.27 (0.90–1.82, p = 0.178) |
| | Third | 407 | 158 (38.8%) | 1.15 (0.86–1.53, p = 0.350) | 1.31 (0.93–1.84, p = 0.123) |
| | Second | 366 | 205 (56.0%) | 2.30 (1.72–3.10, p<0.001) | 2.41 (1.71–3.43, p<0.001) |
| | Poorest | 399 | 252 (63.2%) | 3.10 (2.32–4.16, p<0.001) | 3.85 (2.73–5.46, p<0.001) |
| Household size | 7+ | 1,030 | 427 (41.5%) | Ref | Ref |
| | 1–6 | 882 | 453 (51.4%) | 1.49 (1.24–1.79, p<0.001) | 1.25 (1.01–1.54, p = 0.044) |
| Patient was breadwinner | No | 1,115 | 456 (40.9%) | Ref | Ref |
| | Yes | 797 | 424 (53.2%) | 1.64 (1.37–1.97, p<0.001) | 1.07 (0.83–1.38, p = 0.611) |
| Employment status before TB | Unemployed | 917 | 352 (38.4%) | Ref | Ref |
| | Employed | 995 | 528 (53.1%) | 1.81 (1.51–2.18, p<0.001) | 2.26 (1.74–2.93, p<0.001) |
| Drug resistance status | DS-TB | 1,592 | 593 (37.2%) | Ref | Ref |
| | DR-TB | 320 | 287 (89.7%) | 14.65 (10.23–21.67, p<0.001) | 8.27 (5.27–13.28, p<0.001) |
| Treatment history | New | 1,431 | 517 (36.1%) | Ref | Ref |
| | Relapse | 311 | 230 (74.0%) | 5.02 (3.83–6.64, p<0.001) | 2.14 (1.51–3.03, p<0.001) |
| | Retreatment | 163 | 127 (77.9%) | 6.24 (4.29–9.29, p<0.001) | 2.08 (1.26–3.43, p = 0.004) |
| | Unknown | 7 | 6 (85.7%) | 10.61 (1.81–200.62, p = 0.029) | 6.84 (1.03–134.47, p = 0.086) |
| Mode of treatment | Self-administered | 289 | 93 (32.2%) | Ref | Ref |
| | With treatment partner | 1,616 | 782 (48.4%) | 1.98 (1.52–2.59, p<0.001) | 1.62 (1.20–2.19, p = 0.002) |
| | Other | 7 | 5 (71.4%) | 5.27 (1.11–37.27, p = 0.050) | 5.76 (1.15–42.44, p = 0.045) |
| Hospitalized during the current treatment phase | No | 1,842 | 820 (44.5%) | Ref | Ref |
| | Yes | 70 | 60 (85.7%) | 7.48 (3.98–15.61, p<0.001) | 9.47 (4.65–21.01, p<0.001) |

surveys, only two countries (Ghana and Kenya) so far included BMI in the survey, and they reported that more than 50% of the survey participants had a BMI of less than 18.5 [11, 12]. Further studies assessing low BMI and financial burden from direct non-medical costs are required in the Philippines and Asian contexts.

Income loss also accounted for a large proportion of total TB patient costs (40.4%) as per this survey. TB impoverished the patients' households further. The proportion of households living under the poverty line increased from 68% to 78% during TB treatment. The Philippines has a conditional cash transfer programme for households living in poverty, "Pantawid Pami-lyang Pilipino Program (4Ps)", and the programme was started in 2007 by the Department of Social Welfare and Development (DSWD). The programme covered 4.4 million households as

of 2016 [33]. However, a report pointed out that 4P benefits covered only 60% of the poor households, and 27% of the beneficiaries were non-poor households [23, 34]. The result of our survey also highlighted that only 1.3% of TB-affected households were receiving 4P benefits even though more than three-quarters of the households of survey participants were living under the poverty line and supposed to be eligible for 4Ps. A similar situation was reported for a national disability grant for formal employers in TB patient cost survey in Lao PDR. Although more than 10% of the survey participants were employed in formal sector, and TB patients are eligible to receive disability grant or sickness benefit for 6 months in Lao PDR, only 0.4% reported that they received any types of social welfare [25]. The survey in Lao PDR recommended re-designing the claim mechanism in collaboration with a government agency, which is providing sickness benefit for formal employees, may facilitate wider access to the sickness benefit. Therefore, linking TB patients' information from NTP to DSWD and assessing their eligibility for 4P benefits at TB diagnosis may save TB-affected households from financial catastrophe due to TB.

Even though free TB services are provided under the NTP engaged health facilities in the Philippines, TB patients still incur a substantial amount of direct medical costs (US$101, 16.9% of total costs). Not only did TB-affected households incur direct non-medical costs due to additional supplements or foods, but also they paid for prescribed nutritional supplements (e.g., vitamins other than TB medications) during medical follow-ups as direct medical costs. In the Philippines, as a common clinical practice, patients are prescribed with nutritional supplements to reassure effective treatment and at times faster cure, and prevention of other diseases. Many TB patients also suffer malnutrition. Patients might have attributed the prescription as part of TB cure when this is actually addressing their nutritional status. Nutritional supplements are also a common form of freebie/giveaway in health facilities as this can be easily dispensed to any kind of patient including TB patients. However, a Cochrane systematic review concluded that there is no evidence that micronutrient supplements (e.g., vitamin supplements) impact on treatment completion, sputum conversion, or weight gain for TB patients [35]. The current NTP guideline recommends vitamin prescription (Vitamin $B_6$) only for TB patients who complain of a burning sensation as an adverse reaction due to Isoniazid and for all pregnant or breastfeeding women and infants taking isoniazid [36, 37]. Although further investigation would be required to understand the reasons for the frequent prescriptions of vitamins during follow-up visits, TB practitioner's compliance to the NTP guideline may reduce unnecessary prescriptions of micronutrient supplements for TB patients, which could result in mitigating patients' financial burden as well as a reduction in TB provider costs in the Philippines.

Extensive financial supports are provided for DR-TB patients who seek care in NTP facilities through a TB enabler package, funded by the Global Fund. The proportion of DR-TB patients who received vouchers were 70% in our survey. This was higher than other countries such as Lao PDR (10% of DR-TB survey participants reported received vouchers) [25]. However, the impact of the enabler package on reducing the overall proportion of catastrophic costs was minimal (from 42.4% to 42.0%) as it targets only DR-TB patients, which accounted for 3.5% of estimated incident cases in the Philippines [16]. Expansion of the TB enabler package to DS-TB patients, especially those living under the poverty line, might be required for a substantial reduction in the proportion of catastrophic costs due to TB. Such an enabler package could cover income loss and additional food costs. Also, transport costs need to be reduced by decentralizing TB care from facilities to communities and homes (e.g., using community-based sputum transport, home-based and community-based administration of drugs, digital adherence solutions).

## Strengths and limitations

This survey is the first national TB patient cost survey conducted in the Philippines. Thus, survey results serve as a baseline to monitor the progress towards the goal of the End TB Strategy to eliminate catastrophic costs due to TB. Moreover, the data collection and interviews of this survey were conducted from a total of 1,912 TB patients (the TB Patient Cost Survey with the largest sample size conducted as of date). This allows the presentation of a wide variety of risk factors associated with catastrophic costs with statistical significance (e.g., urban/rural, DR-TB, coping mechanisms, TB treatment history), which cannot be assessed in other TB patient cost surveys due to limited sample sizes [11, 25].

However, this survey had several limitations too. First, designed as a cross-sectional study, an extrapolation method recommended by the WHO had to be used to estimate costs for the entire duration of the TB episode (from the onset of symptoms to the end of TB treatment) [15]. Longitudinal studies, having interviews of each TB patient at multiple time points during their TB episodes, can capture more accurate TB patient costs and catastrophic costs. A recent study, which also assessed TB patient costs in the Philippines, raised an issue that the current survey instrument of TB patient cost surveys may underestimate the cost of facility visits and costs for the pre-diagnosis period [38]. Thus, conducting a longitudinal study can help identify necessary improvements in the current survey protocol and instrument. Furthermore, this survey only included those who initiated TB care at NTP-engaged health facilities, therefore, results might underestimate TB diagnostic delay and related pre-diagnostic costs. Second, the survey only included TB patients who are currently on treatment at the time of data collection. Thus, patients who tested positive for TB but never initiated TB treatment and those who had adverse treatment outcomes before the survey and those who were taking TB preventive treatment were not covered by this survey. Third, although this survey enrolled TB patients from more than 188 health facilities to ensure national representativeness, all of them were NTP-engaged health facilities. Therefore, similar to other TB patient cost surveys conducted so far, financial burden and catastrophic costs faced by patients receiving TB treatment in private health facilities could not be assessed in our survey. It is well-known that many TB cases are diagnosed and treated by private service providers, and a majority of them are unreported to the NTP [39, 40]. In the Philippines, a TB patient pathway analysis presented that one-third of TB patients sought their initial care in the private sector, and only 9% were notified from private facilities in 2015 [41]. Recently, the Philippines government implemented mandatory TB case reporting from private facilities, and the proportion of TB case notifications from the private sector has been rapidly increasing (25% in 2018) [42, 43]. It would be worth including private health facilities in the second TB patient cost survey to understand the financial burden incurred by TB patients treated in the private sector. Fourth, following the WHO recommended method, the survey investigated patient costs from the onset of TB symptoms until the end of TB treatment. Therefore, costs and prolonged social and economic consequences after completing TB treatment were not part of the survey (e.g., permanent job loss).

## Conclusion

TB patients in the Philippines are facing a substantial financial burden (US$ 601) due to TB despite free TB services provided under the National TB Programme. 42% of TB-affected households faced catastrophic costs due to TB. The current TB enabler package mitigates catastrophic costs to some extent, but only for DR-TB patients. Expansion and enhancement of the current social and welfare support through multi-sectoral collaboration is urgently required to achieve zero catastrophic costs due to TB in the Philippines.

## Supporting information

**S1 File.**
(DOCX)

## Acknowledgments

This article is anchored on the TB patient cost survey realized through a collaborative effort among the DOH-NTP, WHO Philippines, USAID support through the Health Policy Development Program managed by UPecon Foundation Inc. and The Global Fund Grant for TB in the Philippines with Philippine Business for Social Progress as its principal recipient. This article's contents are the responsibility of the authors and do not necessarily reflect the views of the sponsors.

## Author Contributions

**Conceptualization:** Anna Marie Celina Garfin, Donna Mae G. Gaviola, Rajendra Prasad Yadav, Andrew Siroka.

**Data curation:** Jhiedon L. Florentino, Rosa Mia L. Arao.

**Formal analysis:** Jhiedon L. Florentino, Rosa Mia L. Arao, Takuya Yamanaka.

**Funding acquisition:** Anna Marie Celina Garfin, Donna Mae G. Gaviola, Rajendra Prasad Yadav.

**Investigation:** Jhiedon L. Florentino, Rosa Mia L. Arao, Donna Mae G. Gaviola, Carlos R. Tan.

**Methodology:** Jhiedon L. Florentino, Rosa Mia L. Arao, Carlos R. Tan.

**Project administration:** Donna Mae G. Gaviola.

**Supervision:** Jhiedon L. Florentino, Anna Marie Celina Garfin, Donna Mae G. Gaviola, Carlos R. Tan, Andrew Siroka, Nobuyuki Nishikiori.

**Validation:** Jhiedon L. Florentino, Rosa Mia L. Arao, Takuya Yamanaka.

**Visualization:** Takuya Yamanaka.

**Writing – original draft:** Jhiedon L. Florentino, Rosa Mia L. Arao, Donna Mae G. Gaviola, Carlos R. Tan, Takuya Yamanaka.

**Writing – review & editing:** Rosa Mia L. Arao, Anna Marie Celina Garfin, Donna Mae G. Gaviola, Carlos R. Tan, Rajendra Prasad Yadav, Tom Hiatt, Fukushi Morishita, Takuya Yamanaka, Nobuyuki Nishikiori.

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
