## [Decision Letter · Decision Letter 0]

17 Dec 2021

PONE-D-21-12327Expansion of social protection is necessary towards zero catastrophic costs due to TB: The first national TB patient cost survey in the PhilippinesPLOS ONE

Dear Dr. Arao,

Thank you for submitting your manuscript to PLOS ONE. After careful consideration, we feel that it has merit but does not fully meet PLOS ONE’s publication criteria as it currently stands. Therefore, we invite you to submit a revised version of the manuscript that addresses the points raised during the review process.

The reviewers have suggested a lengthy list of suggestions, mainly to make the paper easier and clearer to read, as well as being more precise on methods and on presentation of results. I hope this will further improve a paper that both reviewers feel is interesting and of a high standard.

We look forward to receiving your revised manuscript.

Kind regards,

Susan Horton

Academic Editor

PLOS ONE

Journal Requirements:

2. Thank you for stating in the text of your manuscript "The St. Cabrini Medical Center- Asian Eye Institute Ethics Review Committee (SCMC-AEI ERC) reviewed and provided a national ethics approval for the study protocol and survey instrument (ERC #2016-020). Individual ethics approvals were also obtained from Baguio General Hospital and Medical Center Ethics Review Committee (BGH-MC ERC), Department of Health XI Cluster Ethics Review Committee (DOH XI CERC), and the National Children's Hospital Institutional Review Board (NCH-IRB) for the three cluster facilities with their own Internal Review Board (IRB). A written consent form was obtained from all participants before the commencement of the interview. For participants less than 18 years old, we obtained written consent from their parents or legal guardians who accompanied them." Please also add this information to your ethics statement in the online submission form.

3. Please provide additional details regarding participant consent. In the ethics statement in the Methods and online submission information, please ensure that you have specified whether consent was informed.

Reviewers' comments:

Reviewer's Responses to Questions

**Comments to the Author**

1. Is the manuscript technically sound, and do the data support the conclusions?

Reviewer #1: Yes

Reviewer #2: Yes

2. Has the statistical analysis been performed appropriately and rigorously? 

Reviewer #1: Yes

Reviewer #2: Yes

3. Have the authors made all data underlying the findings in their manuscript fully available?

Reviewer #1: No

Reviewer #2: Yes

4. Is the manuscript presented in an intelligible fashion and written in standard English?

Reviewer #1: Yes

Reviewer #2: Yes

5. Review Comments to the Author

Reviewer #1: This is an interesting paper detailing the findings of an important study which I hope is published. The authors report the findings of a national tuberculosis patient cost survey conducted in Philippines, the largest such study in any country to date. While the paper and analysis are generally of a high standard, there are some issues that require attention prior to publication.

General:

• Tables of results are presented split by Rural DS patients, Urban DS patients and DR-TB patients with p-values from statistical tests provided in a final column. Lines 125-127 state: ‘Statistical differences between urban and rural DS-TB patients and DR-TB patients were examined using a chi-square test for categorical data and the Welsh T-test or two-sample Wilcoxon rank-sum test for continuous data’. This statement is ambiguous, as Welch’s t-test and two-sample rank-sum test are tests for two samples rather than three. It seems possible that the authors may have been suggesting that they had ‘lumped’ the two DS-TB populations together (Urban and Rural), testing for significant differences between values in this combined population to the DR-TB population. If this is indeed the case, statements such as ‘…food insecurity was higher in rural DS-TB patients (26.1%, p<0.001).’ (line 236-237) are not easily interpreted and potentially misleading. This issue requires clarity. Furthermore, (assuming DS-TB populations have been lumped) it would be useful for the reader to also be provided with results of tests to know if differences observed between DS-TB urban and DS-TB rural were significantly different. For example, 44% of urban DS-TB patients had a delay before diagnosis vs 53% of rural; 50% of rural DS patients are reported to have health insurance, while this is 33% in rural DS patients; 45% of Rural DS patients employed a coping mechanism vs 35% of Urban DS patients etc. Perhaps having two columns rather than one, showing the p-value for DS-urban vs DS-Rural and (what’s assumed to be) the current combined DS vs DR would be worthwhile. This would be particularly valuable for understanding differences in time lost in Table 4.

• The paper highlights a very important point in the Methods section. Namely, that there are assistance programmes available to tuberculosis patients in Philippines which data have suggested are not being utilised. Lines 37-42 detail a national health insurance scheme (PhilHealth), with population coverage of over 90% in 2019, which for TB is said to cover costs for ‘diagnosis, treatment, reporting, drugs, consultation, and health education’. The authors then add —likely surprising to many readers — that (according to a WHO report – ref [21]) only 11% of notified TB patients in Philippines make any claims. The findings of the present study further underline this important under-utilisation of social protection schemes, finding ‘only 1.3% of TB-affected households were receiving 4P benefits even though more than three-quarters of the households of survey participants were living under the poverty line and supposed to be eligible’. While the authors offer a sentence suggesting that patient information from the NTP could be linked to the Department of Social Welfare and Development (lines 314-316), it seems important that more discussion is devoted to this issue, briefly considering why patients may not be accessing this support, ideally with reference to other studies which might illuminate this issue. This is particularly relevant as the authors suggest that the Global Fund enablers package (which they demonstrate had a considerably more successful uptake of 70% with DR-TB patients) could be expanded to DS-TB patients (line 341), however even if available for DS-TB patients, can there be confidence that this additional offer would be utilised, when other available benefits are not?

• Additional clarity is needed on nutritional costs earlier in the paper, as currently the lack of early distinction could lead to confusion. The authors detail in the discussion that there are two types of nutritional/additional supplements (and go on to have a useful discussion about these supplements).

‘Not only did TB-affected households incur direct non-medical costs due to additional supplements or foods, but also they paid for prescribed nutritional supplements (e.g., vitamins other than TB medications) during medical follow-ups as direct medical costs.’ (lines 319-320)

While this distinction is helpfully made clear in the Discussion, this is not clear earlier on in the paper. The Methods list only one type of ‘nutritional and food supplements’, classifying these under medical costs:

‘Direct costs include out-of-pocket expenditures for TB care such as medical expenses for diagnosis and treatment (hospitalization, consultation fees, radiography, laboratory test, other procedures, medicines, and nutritional and food supplements)…’ (lines 87-90)

While examples of non-medical costs are given as:

“non-medical costs (transportation, food, and accommodation expenses for patient and companion)” (Lines 89-90).

It would be helpful if the authors included an earlier mention of nutritional supplements as non-medical costs in the Methods, especially considering they find ‘…costs for additional food and supplements other than regular diets comprised a large proportion among components of direct non-medical costs’ (lines 205-206).

• Line 68: The instrument used for data collection has not been provided, however it is stated that ‘A locally adapted version of the World Health Organization (WHO) protocol and patient cost survey instrument’ was used. It would be useful for the authors to here provide readers with a reference for the WHO patient cost survey (and perhaps either make the adapted instrument they used available online or make clear can be accessed on request). Presumably the survey was asked in Tagalog, however this should also be specified.

• The method for estimating pre-treatment/pre-diagnostic costs are not detailed in the text – one sentence covering this in the methods would be beneficial. Additionally, while the authors mention in their Discussion that there have been issues identified elsewhere with TB patient costs surveys underestimating pre-diagnosis costs (line 359-361), it would be useful for the authors to comment specifically on the impact of this in their own study, which found only 3.2% of costs attributable to the pre-diagnosis phase (Fig 1), despite the median time to a diagnosis being 3.9 weeks (line 178), and half of patients reporting a delay in diagnosis (line 176). This could perhaps also be linked to the issue stated in the Discussion that ‘one-third 371 of TB patients sought their initial care in the private sector’ (line 370-371). While it’s not given what proportion of these patients end up in the public sector, one would expect this to inevitably drive-up pre-diagnostic costs/visits, which appear underestimated here.

• Line 79: ‘Consultations with NTP managers and health workers also helped verify the reported average expenses on various expense items’. It is not clear at what point in data collection these consultations took place, if any data were altered as a result, or — if they happened during data collection — whether methods were altered. Some additional information to explain this is required.

Minor:

• Phrases appear in tables which are not explained or defined in the text: ‘working hour basis’ (Table 4), ‘output approach’ (Table 5). These should be explained briefly in the text.

• The definition for catastrophic cost is given three times in the paper (lines 20-21; 84-86; 133-134). Suggest one of these definitions in the Methods is removed to avoid unnecessary repetition.

• While the number of patients required under the authors’ sample size calculations were met, the authors do not make clear the response rate for the survey – if available, this should be included.

• Headings for Tables, 2-7 should be revised to remove the unnecessary ‘…of the National TB Patient Cost Survey, Philippines 2016-2017’ or equivalent.

Abstract:

‘The mean total cost of TB treatment was US$601 overall’ please rephrase to be clear that this also includes pre-diagnostic costs.

‘This was nearly five times higher among DR-TB patients, estimated at $US2,919’. This is when compared to the weighted mean, which includes the costs of DR-TB patients. Costs incurred by DR-TB patients were in fact over 5 times larger than those incurred by DS-TB patients, and it would be worthwhile for the authors to rephrase to make this clear.

‘Overall, 42.4% (…) of TB-affected households would have faced catastrophic costs due to TB’ – could the phrasing ‘would have faced’ be adjusted to ‘faced’?

Introduction:

Line 11 ‘The total cost of TB as a proportion of the TB patient’s household income…’ – could this be rephrased as ‘Total TB-related costs incurred by patients as a proportion of their annual household income…’ for clarity.

Methods:

Line 32 – requires a reference for national income per capita

Line 41-42 – reference required for Department of Social Welfare’s cash transfer program for poorest families.

Line 44 ‘using Global Fund grant’ � ‘using a Global Fund grant’

Line 63 ‘a total of 188 BMUs was randomly selected ’ � ‘…were randomly selected…’

Line 70: ‘Patients over 18 were the main respondents.’ – suggest rewording to ‘Most patients interviewed were 18 or over.’ (or provide percentage) to be clear that the authors are not suggesting that the likelihood of a given patient participating was related to their age (unless this was the case?).

Line 97: ‘Moreover, to avoid recall bias…’ suggest rewording to ‘reduce recall bias’.

Line 99: Should this ‘or’ be an ‘and’?

Line 113-114 states: ‘Indirect cost was estimated as income loss, which is the difference between the family income of all members before and during the TB illness’. Table 1(line 119) however does not include income in the ‘before treatment’ column (the two categories of ‘before treatment’ data listed as having been collected are ‘medical’ and ‘non-medical’ suggesting only direct cost data were collected, and only for new patients interviewed in the IP). If not included here in the table, perhaps a footnote clarifying that pre-treatment income had been collected for all patients would be beneficial to the reader.

Line 126: Welsh T-test � Welch’s t-test

Line 129: ‘…at the rate of php 50.11 per US$ 1.’ With data having been collected in 2016-2017, it would be useful if the authors could indicate the timing of this exchange rate, ideally providing a reference.

Results:

Table 2: Line 170: Household size category of 0-6. Unclear what a household size of ‘0’ would mean – should this be ‘1-6’?

Line 174: ‘About half of the participants (52.8%) were bacteriologically confirmed’. This is repeated and covered in more detail in lines 182-184. Suggest removing from line 174.

Line 178: Delete repeated ‘in visits’

Table 3 (line 190): ‘Hospitalised during current phase’ – a footnote here (or a mention in the Discussion) would be useful to highlight that this percentage likely under-represents the true level of hospitalisation among patients as they were sampled cross-sectionally.

Line 197: ‘(…rural DS-TB 104.6 hours, p<0.001…);’ - The p-value provided here in the text of <0.001 is different to that provided in Table 4 (0.027).

Line 192; Line 194, Table 4: The word ‘care giver’ is used here while ‘companion’ is used elsewhere. Suggest using a single term throughout and adding a note somewhere to clarify difference between this person and their ‘treatment partner’.

Line 212 – suggest replacing word ‘heavier’ with ‘greater’ or similar.

Line 233: ‘About one-third of TB-affected households (29.1%) lost their jobs…’ - this statement needs rewording

Discussion:

The Discussion is generally of a good quality, covering interesting points.

Line 284: ‘As of July 2020…’ Could this be updated?

Line 295: ‘…5 of 14 countries…’ – Line 284 mentions 19 countries – unclear what the 14 here refers to.

Lines 327-328 The findings of the Cochrane review quoted by the authors: ‘a Cochrane systematic review concluded that there is no evidence that micronutrient supplements (e.g., vitamins) affect … weight gain for TB patients’ appears to contradict the earlier statement on lines 296-298: ‘TB patients usually require an increased intake of foods and nutrients to recover from TB-associated weight loss and malnutrition’. Suggest that this earlier statement on lines 296-298 is amended, perhaps to reflect that this is a common belief, rather than presenting as a statement of fact.

Line 352: TB-PCSs – this needs writing in full

Conclusion:

Line 384: ‘…would be urgently required…’ - suggest change to ‘…is urgently required…’.

Reviewer #2: Dear Dr Arao and Colleagues,

Many thanks for the opportunity to review this manuscript, which describes the economic burden of TB-related cost on TB affected families in the Philippines and adds to the literature from the Western Pacific region and globally by providing a specific focus on the Philippines.

The manuscript is well-written and I do not have major comments but I would recommend a careful review of the language to improve readability of the manuscript. My comments are listed below.

Line 16: please replace pillars with targets

Line 21: please specify pre-TB annual household income

Line 23: it should be on instead of in

Line 24: I would use the past: aimed

Line 33: served but I believe they still are?

Line 133-134: repetition of definition of catastrophic costs which also features earlier in the manuscript. Please consider consolidating this as well as the use of acronyms (e.g. WHO)

Line 153: please consider using of these instead of those

Line 229: living with less than. With is missing

Line 248: “This indicated the proportion of TB patients would”; “that” is missing

Line 284: I would say “implementation of national TB patient cost surveys”

Line 309: suggest linking up sentences to improve readability

Line 328: instead of affect, I would say have an impact on

6. PLOS authors have the option to publish the peer review history of their article (what does this mean?). If published, this will include your full peer review and any attached files.

Reviewer #1: No

Reviewer #2: No

---

## [Author Response · Author response to Decision Letter 0]

5 Feb 2022

1. Thank you for providing a Data Availability Statement and explaining why the data are restricted. In order for your statement to meet our data sharing policy requirements (https://journals.plos.org/plosone/s/data-availability), however, we ask that you provide the following information:

A. The institutional body who has imposed the restrictions upon your data (e.g., a Research Ethics Committee or Institutional Review Board, etc.)

The St. Cabrini Medical Center- Asian Eye Institute Ethics Review Committee (SCMC-AEI ERC)

B. A non-author contact that researchers can make data access queries to (in addition to the permissions they will have to obtain from the National TB Programme of the Philippines (ntp.mne@gmail.com))

Jose Gerard B. Belimac, MD, MPH 

Team Lead

Infectious Diseases and Adult Health Division (Concurrent) and,

Evidence Generation and Management Division

DOH San Lazaro Compound, Rizal Ave. Sta. Cruz, Manila 1003

email: tuberculosis@doh.gov.ph

Our responses to editor's comments are as follows. 

The responses to all the comments from reviewers are detailed and uploaded in a separate file named "Responses to Reviewers".

Following the guideline, we updated our manuscript and regenerated files for figures using PACE.

2. Thank you for stating in the text of your manuscript "The St. Cabrini Medical Center- Asian Eye Institute Ethics Review Committee (SCMC-AEI ERC) reviewed and provided a national ethics approval for the study protocol and survey instrument (ERC #2016-020). Individual ethics approvals were also obtained from Baguio General Hospital and Medical Center Ethics Review Committee (BGH-MC ERC), Department of Health XI Cluster Ethics Review Committee (DOH XI CERC), and the National Children's Hospital Institutional Review Board (NCH-IRB) for the three cluster facilities with their own Internal Review Board (IRB). A written consent form was obtained from all participants before the commencement of the interview. For participants less than 18 years old, we obtained written consent from their parents or legal guardians who accompanied them." Please also add this information to your ethics statement in the online submission form.

We added the full sentences in the online submission form.

3. Please provide additional details regarding participant consent. In the ethics statement in the Methods and online submission information, please ensure that you have specified whether consent was informed.

We added some more details of informed consent process in our revised manuscript (Line 146-156).

Our survey dataset contains privacy-sensitive information including participant’s individual and household income that formed a core part of the analysis. Even though we remove patient’s identifiers such as patient number and name, there is still a possibility that those who are familiar with the project sites and beneficiaries may be able to identify participants and their households. The informed consent signed by all participants explicitly mentioned that only the research team have access to the data set. Due to such ethical and confidentiality restrictions, the survey dataset will be made available only upon request and with permission from the National Tuberculosis Control Programme (NTP), Department of Health, Philippines. All interested researchers will contact the corresponding author (rosamia.arao@gmail.com) and the National TB Programme of the Philippines (ntp.mne@gmail.com) to request the data access.

We lined the corresponding author’s ORCID ID in the online submission.

We updated reference following this comment and reviewers’ suggestions. We don’t have retracted references in the list.

---

## [Editor Report · Decision Letter 1]

16 Feb 2022

Expansion of social protection is necessary towards zero catastrophic costs due to TB: The first national TB patient cost survey in the Philippines

PONE-D-21-12327R1

Dear Dr. Arao,

We’re pleased to inform you that your manuscript has been judged scientifically suitable for publication and will be formally accepted for publication once it meets all outstanding technical requirements.

Kind regards,

Susan Horton

Academic Editor

PLOS ONE

Additional Editor Comments (optional):

Thank you for the work you did to address the reviewers' suggestions so thoroughly.
---

## [Editor Report · Acceptance letter]

18 Feb 2022

PONE-D-21-12327R1 

Expansion of social protection is necessary towards zero catastrophic costs due to TB: The first national TB patient cost survey in the Philippines 

Dear Dr. Arao:

I'm pleased to inform you that your manuscript has been deemed suitable for publication in PLOS ONE. Congratulations! Your manuscript is now with our production department. 

Kind regards, 

on behalf of

Dr. Susan Horton 

Academic Editor

PLOS ONE